# Laws Governing Nitrogen Loss and Its Numerical Simulation in the Sloping Farmland of the Miyun Reservoir

**DOI:** 10.3390/plants12102042

**Published:** 2023-05-19

**Authors:** Yan Li, Liang Jin, Jiajun Wu, Chuanqi Shi, Shuo Li, Jianzhi Xie, Zhizhuang An, Linna Suo, Jianli Ding, Dan Wei, Lei Wang

**Affiliations:** 1Institute of Plant Nutrition, Resources and Environment, Beijing Academy of Agricultural and Forestry Sciences, Beijing 100097, China; li.yan622@163.com (Y.L.); jinliang19762003@aliyun.com (L.J.);; 2College of Resources and Environmental Sciences, Agricultural University of Hebei, Baoding 071000, China; 3Heilongjiang Province Key Laboratory of Cold Region Wetland Ecology and Environment Research, Harbin University, Harbin 150076, China

**Keywords:** nitrogen, surface flow, subsurface flow, nitrogen simulation model

## Abstract

Surface flow (SF) and subsurface flow (SSF) are important hydrological processes occurring on slopes, and are driven by two main factors: rainfall intensity and slope gradient. To explore nitrogen (N) migration and loss from sloping farmland in the Miyun Reservoir, the characteristics of total nitrogen (TN) migration and loss via SF and SSF under different rainfall intensities (30, 40, 50, 60, 70, and 80 mm/h) and slope gradients (5°, 10°, and 15°) were studied using indoor stimulated rainfall tests and mathematical models. Nitrogen loss via SF and SSF was found to increase exponentially and linearly with time, respectively, with SSF showing 14–78 times higher loss than SF. Under different rainfall intensities, SSF generally had larger TN loss loading than SF, thereby indicating that SSF was the main route for TN loss. However, the TN loss loading proportion via SF increasing from 14.03% to 35.82% with increasing rainfall intensity is noteworthy. Furthermore, compared with the measurement data, the precision evaluation index Nash-Suttcliffe efficient (NSE) and the determination coefficient (R^2^) of the effective mixing depth model in the numerical simulation of TN loss through SF in the sloping farmland in the Miyun Reservoir were 0.74 and 0.831, respectively, whereas those of the convection-dispersion equation for SSF were 0.81 and 0.811, respectively, thus indicating good simulation results. Therefore, this paper provides a reference for studying the mechanism of N migration and loss in sloping farmland in the Miyun Reservoir.

## 1. Introduction

During the utilization of sloping farmland, abundant organic and inorganic fertilizers are often used to increase the output of crops. Since these fertilizers are easily washed away by hydraulic erosion, their absorption and utilization rates are low, which causes nutrient accumulation in the downstream water bodies. Additionally, rainfall erosion destroys the original soil structure, causing the loss of sediment and nutrients along with SF, thus generating severe non-point source pollution [1]. Moreover, soil nutrient loss caused by severe erosion directly threatens food and ecological environment security.

Sustained soil erosion results in the loss of valuable agricultural land and decreased crop yield. With the rapid growth of population, food demand is increasing day-by-day [2,3]. Conserving soil by controlling erosion is crucially important [4]. Therefore, understanding the laws and influencing factors of N loss on the soil of sloping farmland is highly significant for controlling soil and water loss as well preventing agricultural non-point source pollution.

SF and SSF are the main components contributing to runoff on the soil slope. Existing study results show that N loss associated with SSF contributes more to N migration and loss in sloping land, with SSF being the main determining factor. Hence, reducing SSF is vital for controlling N loss [5]. Topography and rainfall are the main natural factors causing loss of soil, water, and nutrients. Topography can be subdivided into factors including slope gradient, slope length, and micro-topography [6,7,8], while rainfall can be subdivided into rainfall intensity, rainfall volume, rainfall duration, raindrop kinetic energy, etc. [9,10] Among these factors, slope gradient and rainfall intensity have recently become the hot spots of research in China and abroad. The time of rainfall infiltration and SF rate are affected by slope gradient. N loss associated with SF significantly increases with increasing slope gradient, with the peak value occurring at a 10°–15° slope gradient, whereas N loss associated with SSF varies with the slope [11]. Furthermore, rainfall intensity generally impacts SF more than SSF, because SF is the first phase and is directly influenced by land cover. SF is more highly erosive than SSF and is more likely to attain high velocities. At low rainfall intensity, the erosion effect on the surface and N loss associated with SF are both small. However, as rainfall intensity increases, nutrient loss is aggravated, with the greatest loss risk occurring in the early stage. N loss associated with SSF increases with increasing rainfall intensity, in which SSF plays a major role [12].

Numerous studies have evaluated the characteristics of runoff generation and the related N and phosphorus (NP) loss to help reduce NP loss and prevent non-point source pollution. Furthermore, some scholars have studied this on different land use types in a certain area. For example, Zhou et al. monitored and studied the SF sediment yield of loess slopes under different vegetation cover types [13], while another study [14] evaluated the characteristics of NP loss associated with SF and SSF on different land types in the Hun-River Basin in Liaoning Province. Wang Guozhong et al. studied the NP loss on different land types in the mountainous areas of southwest Henan Province [15], whereas Lv Ting et al. explored the same along with SF in the catchment area of the Hexi Reservoir in Changxing County based on different land use types [16]. Additionally, some scholars have studied how different soil and water conservation measures have impacted runoff generation and NP loss. Wang Lei et al. studied how hedgerows controlled soil erosion and NP export from sloping farmland [17], while Wang Quanjiu et al. explored the impact of different vegetation cover types on runoff generation, sediment loss, and NP loss on loess slopes [18], and Choi et al. studied the control effect of vegetated filter strips (VFS) on NP pollution [19]. Field observation of runoff plots and artificial rainfall simulation are the most utilized study methods [20,21,22]. However, most studies have focused on the characteristics of the generation of SF and NP loss, with SF being the main research object [23,24]. Unfortunately, the proportion of SF and SSF loss along with the accompanying N loss from sloping farmland has received less attention. Although some studies have revealed that the migration characteristics of SF and SSF were quite different, how NP migration via SSF affects eutrophication cannot be ignored [25].

Numerical simulation of solute migration with SF under rainfall has become relatively mature [26,27,28,29]. In the 1980s, Ahuja [30] proposed the concept of effective mixing depth for the concentration of solutes carried by SF and established an ‘effective mixing depth’ model. Based on that model, Wang Quanjiu and Wang Hui, after accounting for soil infiltration, integrated the Philip infiltration equation [31] in the effective mixing depth model to establish complete and incomplete mixing depth models for solute migration to the surface on loess slopes. These models were verified using bromides as the solute samples and have been widely used in studying NP migration of practical pollutants. Unfortunately, they have rarely been used in the application of studies to sloping farmland. Due to the complex process and mechanism of the generation of SSF, very few numerical simulations of the process of N loss through SSF from sloping farmland exist. Currently, only the denitrification-decomposition (DNDC) model [32] has been improved and applied to the simulation of nitrate N leaching flux in SSF in sloping farmland. Therefore, in this paper, focusing on the dynamic N loss process occurring from sloping farmland to the surface and underground during rainfall, the current effective mixing depth model of the N migration and loss with SF was improved to help simulate N loss in the sloping farmland of the Miyun Reservoir. Furthermore, a numerical simulation of the migration process to the underground and loss occurring via SSF was carried out in combination with the convective-dispersion mathematical model. Thus, this provides a reference for establishing a complete and unified N loss mechanism in sloping farmland.

## 2. Materials and Methods

### 2.1. Test Soil

The test soil used to fill in runoff troughs was taken from the agricultural non-point source pollution prevention and control base in Taishitun Town, Miyun District, Beijing, China (117°6′42.08″ E, 40°32′22.02″ N). To ensure the maximum consistency between the test soil and the field soil layer, undisturbed soil moving was utilized to collect and bag soil in its original place every 5 cm from the ground surface, with a total of 10 layers being collected. The physical properties of soil in each layer were tested separately and then filled to the corresponding layer in the indoor runoff trough. During the filling process, the soil density was controlled to ensure consistency. Herein, a total of 10 layers were compactly filled with soil, respectively, with each layer a thickness of 5 cm to ensure the consistency of the soil bulk density. The soil was then put aside for 3–5 days to restore the soil’s natural characteristics before subjecting it to the rainfall test. The pH was 6.33, while the contents of organic matter, TN, available phosphorus, and available potassium were 9.97 g/kg, 0.448 g/kg, 4.55 g/kg, and 45.9 g/kg, respectively [24]. The methods used to determine the physical and chemical properties of the test soil are as follows, the organic matter, TN, available phosphorus, and available potassium content were determined via the potassium dichromate volumetry–outside heating method, sulfuric acid–catalyst digestion method, sodium hydroxide melting–Mo-Sb colorimetry, and sodium hydroxide melting–flame photometry, respectively. The pH was measured using digital RS485 pH meter (seed studio, China).

### 2.2. Test Devices and Materials

The movable hydraulic variable slope runoff trough used for this test was designed according to the effective rainfall area of the artificial rainfall device and was subsequently produced using a steel plate welding. Its dimensions were 200 cm × 50 cm × 60 cm (length × width × height). Two parallel runoff troughs with flexible slope gradients that could be adjusted from 0° to 30° were used for the parallel tests. The front, left, and right sides of the bottom of the troughs were equipped with extension troughs that were 3 cm high. The extensions had a small-aperture metal fine mesh laid inside to collect SSF, whereas SF was collected by the front catchment trough at the upper part of the soil trough. The rainfall device was equipped with a QYJY-502 portable automatic artificial rainfall simulation system (Xi’an Qingyuan Measurement and Control Technology Co., Ltd., Xi’an, China), with a rainfall height of 5 m and a rainfall uniformity coefficient above 80%, while the rainfall intensity continuously varied from 15 to 120 mm/h (Figure 1).

### 2.3. Experimental Design

The test was conducted at the simulation test base of the Beijing Academy of Agriculture and Forestry Sciences, Haidian District, Beijing, China, from June to October 2022. By combining the grading indices of slope surface erosion intensity specified in the Water Conservancy Industry Standard of the People’s Republic of China (SL190-2007) along with the local actual topography conditions, three slope gradients (5°, 10°, and 15°) were designed in the test. The rainfall intensity was set according to the local annual average rainfall and the difference between the rainstorm grades. A total of six rainfall intensities were prepared (30, 40, 50, 60, 70, and 80 mm/h). A total of 18 effective rainfall tests were conducted using a combination of three slope gradients and six rainfall intensities. The soil runoff troughs were covered with a waterproof cloth before each rainfall test, followed by calibrating the rainfall intensity. After the waterproof cloth was removed, the time taken to reach the target rainfall intensity and uniformity (>90%) was measured. A stopwatch was used to record the generation time of SF and SSF on the slope. The total collection time of the SF and SSF samples was 60 min, with samples collected and runoff flow measured every five min. The soil moisture content was monitored before each rainfall test to ensure the same initial soil moisture content in each test.

### 2.4. Data Acquisition and Analysis

The sampling time and the volume of the collected samples were recorded before the samples were taken back to the laboratory, where they were kept undisturbed to allow sedimentation. Then, the supernatant was poured into a clean polyethylene bottle and stored in a refrigerator at 4 °C for the TN concentration analysis, which was completed within 48 h. The TN concentration was determined via a discrete chemistry analyzer (Smart Chem 200, Alliance, Paris, France).

The loss loading was calculated according to the following equation:(1)$$L=\sum \frac{C_{i}\cdot q_{vi}\cdot t_{i}}{S\times 10^{5}}$$
where, *L* refers to the TN loss loading (kg/hm^2^), *S* is the area of loss (m^2^), *C_i_* is the TN loss concentration at the ith sampling (mg/L), *q_vi_* is the flow at the ith sampling (mL/s), and *t_i_* is the time interval between two samplings (s).

The values of loss concentration and loss loading were calculated using Microsoft Excel, while the data charts were drawn by Origin. The significance analyses of the TN loss through SF and SSF were conducted at a 95% confidence interval (*p* < 0.05) by the least significant difference (LSD) method in the Statistical Package for the Social Sciences (SPSS) software.

### 2.5. Numerical Simulation

In this paper, the mathematical model constructed and evaluated based on the test data mainly comprised two parts: (1) the upward migration of solutes from the soil surface to SF and (2) downward migration of solutes to the underlying soil and loss associated with SSF.

#### 2.5.1. Theoretical Model

##### Model of Simulating N Loss through SF

For the numerical simulation of N migration and loss through SF in the sloping farmland, the effective mixing depth model was used:(2)$$C\left(t\right)=C_{0}exp [-\frac{\left(t-t_{p}\right)R}{h_{m}\left(\theta_{s}+\rho_{s}k\right)}]$$
where, *h_m_* stands for the effective mixing depth (cm), *C* is the lost solute concentration (mg/L), *C*_0_ is the solute concentration initially migrating to SF (mg/L), *θ_S_* is the saturated water content (cm^3^/cm^3^), *ρ_s_* is the soil bulk density (g/cm^3^), *K* is the soil adsorption coefficient (cm^3^/g), *R* is the rainfall intensity (cm/min), *T* is the rainfall time (min), and *T_p_* is the runoff generation time on land surface (min).

Ahuja [30] found that the effective mixing depth increased with time, whereas the rate of increase decreased with time. A previous study found a trend in depth change of the mixing layer [33] with ^32^P and bromides taken as test materials. Herein, the continuous raindrops hitting the soil surface gradually increased the mixing depth, until a sealed water layer was formed on the soil surface after rainfall had formed stable runoff on it. This prevented the mixing depth from increasing, thereby reducing the increase rate. Therefore, the effective mixing depth was improved in this study, thereby allowing the establishment of an effective mixing depth model that is consistent with the changes previously proposed by Ahuja [30].
(3)$$h_{m}=h_{0}+h_{n}In\left(\frac{t-t_{p}}{t^{,\;}}+1\right)$$
where, *t’* stands for the rainfall duration (min), *h*_0_ is the initial mixing depth (cm), and *h_n_* is the basic mixing parameter (cm). By substituting Equation (3) in Equation (2), a revised concentration model of the solutes migrated and lost with SF was obtained.
(4)$$C\left(t\right)=C_{0}exp\left\{-\frac{\left(t-t_{p}\right)R}{ [h_{0}+h_{n}In\left(\frac{t-t_{p}}{t^{,\;}}+1\right)]\left(\theta_{s}+\rho_{s}k\right)}\right\}$$

##### Model of Simulating N Loss through SSF

A numerical simulation based on the traditional convection-dispersion mathematical model was carried out here with respect to the migration process of solutes in the soil. The equation is as follows:(5)$$\frac{\partial \theta_{c}}{\partial_{t}}=\frac{\partial }{\partial r_{i}}\left(\theta D_{ij}\frac{\partial_{c}}{\partial_{r_{i}}}\right)-\frac{\partial q_{i}C}{\partial r_{i}}$$
where, *θ* denotes the volumetric soil water content (cm^3^/cm^3^), *t* denotes the rainfall time (min), *D_ij_* denotes the dispersion coefficient (cm^2^/min), *C* denotes the soil TN mass concentration (mg/cm^3^), *Q_i_* denotes the water flux (cm/min), and *R_i_* denotes the spatial coordinate (*i* = 1, 2, *r*_1_ = *x*, *r*_2_ = z, *D*_11_ = *D_xx_*, *D*_12_ = *D_xz_*).

For the Equation (5) solution, a partial differential equation and the finite element meshes with the same scale as the tests were constructed on the HYDRUS-2D software. The convective-dispersion equation of TN migration in the soil was numerically calculated via the physico-chemical equilibrium transfer module nested by HYDRUS-2D, with an observation point correspondingly set at the SSF outlet to obtain the solute concentration carried by SSF. The parameter values in the calculation process are shown in Table 1. More specifically, *ρs* was the soil bulk density (1.16 g/cm^3^). *θ_s_* and *θ_r_* were given initial values by the Rosetta model according to the measured soil particle distribution, followed by adjustment and correction of the simulation process. *k* was determined by applying the linear isothermal adsorption method. *DL* and *DW* indicate the longitudinal dispersion of TN and the diffusion coefficient in free water, respectively, which were determined through backward deduction based on the simulation results.

#### 2.5.2. Verification and Evaluation of the Model

In addition to the visual comparison between the measured values and the simulated values using graphics, three precision indices: (1) the mean absolute error (MAE), (2) root mean square error (RMSE), and (3) Nash–Suttcliffe efficient (NSE), were used to evaluate the precision of the simulation results. The equations are:(6)$$MAE=\frac{\left(\sum_{i=1}^{n}\left|P_{i}-Q_{i}\right|\right)}{n}\;RMSE=\sqrt{\frac{\sum_{i=1}^{n}{\left(P_{i}-Q_{i}\right)}^{2}}{n}}\;NSE=1-\frac{\sum_{i=1}^{n}{\left(Q_{i}-P_{i}\right)}^{2}}{\sum_{i=1}^{n}{\left(Q_{i}-\bar{Q}\right)}^{2}}$$
where, *n* is the number of samples, *i* = 1, 2,..., *n*. Additionally, *P_i_* refers to the simulated value, *Q_i_* stands for the measured value, and *Q* is the mean value of the measured value. Among them, the optimal MAE and RMSE values are 0, while the optimal value of NSE is 1.

## 3. Results

### 3.1. Characteristics of TN Loss through SF

Figure 2 shows that the TN concentration that migrated with SF under different rainfall intensities and slope gradients exponentially increased with time, along with a sharp increase at the initial stage of runoff generation, followed by stabilization later. When rainfall intensity was ≤40 mm/h, the TN concentration in SF decreased with the increasing slope gradient, which was mainly due to the small rainfall intensity and insufficient erosion force on the soil, thereby resulting in TN being slowly dissolved and released. Additionally, although the increase in SF enhanced the erosion force, it also led to shorter contact between SF and the soil. Although this did not increase the TN concentration, it instead slowly declined. However, when the rainfall intensity was set to >40 mm/h, the TN concentration in SF increased with the increasing slope gradient. This was mainly because TN was adsorbed on the surface of soil particles during the early stage of SF and with the passage of rainfall time, SF gradually increased and fully interacted with the soil, thereby increasing the TN loss. Then, with the TN loss from the surface soil, the TN content in SF gradually decreased and finally steadied with the SF stability. Furthermore, at the same slope gradient, the rainfall intensity was larger and the TN concentration was higher in SF. The reason is that the soil moisture content at the initial stage of rainfall was low, which caused greater rainwater absorption by the surface soil, thereby generating a small SF. As the surface soil moisture content became saturated, SF gradually increased, generating larger soil erosion intensity and ultimately higher TN loss.

### 3.2. Characteristics of TN Loss through SSF

Figure 3 reveals that the error value of the measured TN concentration carried by SSF was much higher than that in Figure 2, thus showing the complexity and uncertainty of the SSF process. The variation of TN loss concentration in SSF with time under different rainfall intensities and slope gradients is shown as follows: (1) In the early stage of SSF, TN concentration first slightly increased and then decreased, which was due to the small flow and the high concentration of free TN in soil. (2) The curves in the middle stage of SSF showed an upward trend until it stabilized at a later stage. With the rainfall intensity ≤40 mm/h, the TN concentration in SSF generally increased with the increasing slope gradient. This happened because a larger slope facilitated water flow and enhanced the contact between water and soil particles, thereby dissolving and causing loss of N fixed on the surface of soil particles, which ultimately created a relatively high TN concentration in SSF. Contrastingly, when the rainfall intensity was >40 mm/h, the TN concentration in SSF generally decreased with the increasing slope gradient, mainly because the great rainfall intensity increased the SSF, which caused the dilution effect.

However, with the same slope gradient, the TN concentration in SSF generally increased with the increasing rainfall intensity, which was because: (1) SSF played a leading role when the rainfall intensity was small. With greater rainfall intensity, greater SSF was generated, which accelerated N loss, and dissolved and moved the non-free N in the soil more quickly. Therefore, the greater the rainfall intensity, the earlier the fluctuation of the concentration curve of TN loss through SSF would start. (2) After a period of SSF generation, the generation of SF lasted for 60 min and the rainfall stopped, which cut off the rainwater supply at the upper part of the SSF, thereby reducing the SSF accordingly, and ultimately increasing the TN loss concentration in the relative SSF.

### 3.3. Comparison of TN Loss between SF and SSF

As shown in Table 2, the TN concentration carried by SSF under the same treatment was significantly higher (*p* < 0.05) than that of TN loss through SF. More specifically, the TN loss concentration with SSF under each treatment was 14–78 times higher than that of SF, which indicated that the TN loss on the slope of sloping farmland was mainly caused by SSF. Under different rainfall intensities and slope gradients, the TN loss through SSF accounted for 87.03–99.86% of the total TN loss in the runoff, which was due to the dual impact of runoff and the associated soil area contact. However, the TN loss through SF accounted only for 0.13–12.97% of the total amount. Therefore, it can be seen that during the implementation of prevention and control activities of agricultural non-point source pollution on sloping farmland, the control of SSF generation, and consequently the soil TN loss, is crucial in reducing the N loss on the sloping farmland. Under different slope gradients, the TN loss proportion through SSF to that through runoff increased with the increasing rainfall intensity, with the maximum being reached at 80 mm/h. This was because the increasing rainfall intensity had a greater impact on SF than SSF, which increased the SF rate, thereby causing greater erosion effect on the surface soil on the slope, ultimately resulting in more N being taken away by SF. Therefore, under different rainfall intensities, no obvious law was observed on the impact of slope factors on the TN loss proportion through SF to that through runoff.

### 3.4. TN Loss Loading of SF and SSF

Figure 4 shows that the cumulative TN loss loading of SF and SSF increased linearly over time. The larger the rainfall intensity, the larger the TN loss rates with SF and SSF. In combination with the average loss loading under each treatment in Table 2, the main migration paths of TN concentration under different treatments involved significantly greater loss loading via underground loss than through the surface (*p* < 0.05). Under the rainfall intensity of 30–80 mm/h, SSF had greater N loss loading than SF. However, with the rising rainfall intensity, the TN loss loading proportion via SF increased from 14.03% to 35.82%. This indicated that with the rising rainfall intensity, N loss was more likely to be via SF.

### 3.5. Numerical Simulation of SF and SSF

As seen in Figure 2, the change of the simulated values of the TN concentration migrating with SF at small rainfall intensities was consistent with the measured values, with the determination coefficient (*R*^2^) reaching 0.9539, 0.9015, and 0.9480, respectively. The Nash–Suttcliffe efficient NSEs were 0.73, 0.75, and 0.95, respectively, at the slope gradients of 5°, 10°, and 15°, with a rainfall intensity of 30 mm/h (Table 3), thereby indicating that not only were the simulation results good, but also the MAE and RMSE values were within a reasonable range. However, the precision of the simulation results decreased with the increasing rainfall intensity, which was mainly because the measured initial loss concentration under great rainfall intensities was small and the TN concentration rapidly stabilized, thus causing an insignificant concentration increase in the actual measurement. Therefore, there was a large error between the exponential decline trend simulated by the model and changes in the measured values. A negative NSE value denoted a poor simulation effect. Nevertheless, NSE was only a part of the evaluation of the simulation results and the MAE and RMSE values and *R*^2^ should be considered. At the rainfall intensities of 30 and 40 mm/h, the MAE and RMSE values were close to the optimal value of 0, while *R*^2^ reached 0.8 and 0.5, respectively, which indicated that the simulation results were acceptable. The simulated and measured values of the surface loss concentration are shown in a scatterplot (Figure 5a). The linear fitting relationship line between the two was *y = 1.0656x + 0.2871* and *R*^2^ was 0.83 after linear regression, which was very close to the 1:1 line. Additionally, MAE, RMSE, and NSE were 0.95 mg/L, 1.54 mg/L, and 0.74, respectively. In general, the revised effective mixing depth model presented good simulation results when simulating N migration and loss with SF in the sloping farmland in the Miyun Reservoir.

Figure 3 showed that the change in trend of the simulated values of the TN concentration loss process carried by SSF under each treatment generally matched the measured values. However, the *R*^2^ value fluctuated greatly (minimum of 0.21 and a maximum of 0.76), which was related to the large error between the measured values and simulated values. The *NSE* values shown in Table 3 also demonstrated fluctuations between positive and negative values under various rainfall intensities and slope gradients, thereby suggesting that the simulation results of TN concentration carried by SSF were as complex and uncertain as the actual loss process. According to the linear regression of the scatter plot of the simulated and measured values of TN concentration carried by SSF shown in Figure 5b, *y = 0.8257x + 10.766* was obtained and *R*^2^ was 0.81, which was also very close to the 1:1 line, with MAE, RMSE, and NSE being 24.99 mg/L, 39.25 mg/L, and 0.81, respectively, thus indicating a good fitting result. Although the MAE and RMSE values of the underground loss simulation results were >25 times those of the surface loss, the error values under each treatment (Table 3) were considered to be within the permissible range, as the TN concentration of groundwater loss was 14–78 times higher than that of surface loss. By comparing the simulation results of TN loss through SF and SSF under various rainfall intensities and slope gradients, the simulation precision on the surface was found to decrease with the increasing rainfall intensity, while the precision underground fluctuated and was uncertain. In conclusion, the simulation precision was satisfactory.

## 4. Discussion

Based on the measured data of the rainfall tests and the numerical simulation results, TN concentration migrating with the SF on sloping farmland was found to exponentially increase with time, thereby indicating that TN concentration carried by the SF increased and stabilized during the process from the SF generation to stabilization. The main reason was that during the process of rainfall generating SF on the slope, the soil surface water content became saturated at the initial stage, allowing it to carry the solutes in the soil to a large extent within the mixing depth, ultimately causing its loss. When the SF was stabilized and the soil surface water content reached saturation, a closed isolation layer was formed between the SF and the soil surface [34], which weakened the solute migration to the SF within the mixing depth of the soil surface. This was similar to previous study results [31,35]. The larger the slope gradient, the shorter time the rainfall stayed on the soil surface, and the smaller the concentration of solutes carried by SF. The larger the rainfall intensity, the more quickly the rainwater would fill the depressions and form a sealed water layer on the soil surface, thus preventing the solutes in the mixing depth from migrating to the SF. In terms of the numerical simulation of N migration with the SF in sloping farmland, an effective mixing depth model with time growth was built based on previous studies [30]. The results showed that the simulation results could well fit the measured data. In this study, the TN concentration error in SF was very small, which enhanced the reliability of the numerical simulation results. Yang et al. [36] found that the simulation results of the effective mixing depth model for N were worse than those of potassium and phosphorus, with the determination coefficient between the TN simulation value and the observed data was only 0.57, which was worse than the simulation results in this study (R^2^ = 0.8308).

As compared with the process of TN loss carried by SF, those with SSF varied significantly between repeated treatments, with no uniform law being observed with the changes in the rainfall intensity and slope gradient, owing to the complexity of the SSF production process and the uncertainty of rainfall infiltration and migration in sloping farmland [37,38]. In this test, an increasing trend of the TN concentration in the SSF under different treatments was presented, which indicated that TN was lost from the soil with water accumulation and migration. On one hand, the solubility and extremely strong mobility of N played a role in it [39], while on the other hand, rainfall had gradually infiltrated into the soil to leach the TN, which ultimately resulted in the gradually increased TN concentration in the underground flow. Although the numerical simulation of the process of TN loss carried by the SSF was simulated by the traditional convection-dispersion equation used to study solute transport and was solved via the HYDRUS-2D software, the simulation results were poor when compared with those about the surface, which was related to the errors between the simulated and measured data. Moreover, some parameters, such as the molecular diffusion coefficient could not be measured in the simulation, as it required backward deduction of the simulation results to obtain appropriate parameter values [40,41]. Therefore, the basic loss law could only be revealed by the simulation results.

In this test, there was no uniform law regarding the impact of slope gradient on TN loss, which matched the previous study results [42]. The difference in TN loss between the SF and SSF has a certain relationship with their generation characteristics. The rate of SF was significantly larger than that of SSF, which resulted in greater SF being generated than SSF at the same time. That diluted the concentration of surface solutes [43], causing the TN concentration carried by the SF to be far lower than that by the SSF. As far as the SSF was concerned, the N concentration carried was high due to a small flow, with the TN being lost mainly through the water transport in the soil. Previous studies [44,45] showed that the SSF in sloping farmland is more likely to increase under small rainfall intensity, resulting in TN concentration carried by the SSF under a small rainfall intensity to be larger than that lost under large rainfall intensity. Under small rainfall intensity, the rate and flow of the SF were also small, thereby leading to small TN loading and a small loss rate. Conversely, the SSF rate and the TN concentration carried by it under small rainfall intensity were large, which increased both the loss loading and loss rate. Hence, the TN loss loading on the soil surface was mainly controlled by the SF flow, while that carried by the SSF mainly depended on the loss concentration.

Therefore, in this study, the change process of the TN concentration migrating and being lost via SF and SSF was simulated numerically. Nevertheless, the generation processes of both could not be simulated by the numerical model, resulting in the nutrient loss loading not being simulated and verified numerically. Moreover, there is a lack of coupling of the numerical model of the surface and underground nutrient loss process. Thus, greater attention should be paid to the above two problems in the future, to support the establishment of a complete and unified nutrient loss mechanism of the sloping farmland.

## 5. Conclusions

A noticeable exponential upward trend with time was observed for the concentration of TN migration and loss with SF in the sloping farmland in the Miyun Reservoir, with a sharp increase seen at the initial stage of runoff generation and a stable trend at the later stage. Under all treatments, the TN concentration lost with the SSF increased with time, with a high error value between repeated treatments. In addition, the TN concentration lost through the SSF was 14–78 times higher than that through the SF, with the linear growth trends with time shown by the TN loss loading through the surface and underground.

At a rainfall intensity of 30–80 mm/h, TN was mainly lost through the SSF. However, with the increasing rainfall intensity, the proportion of TN loss loading through SF rose from 14.03% to 35.82% (i.e., with the rising rainfall intensity, N was more likely to be lost through SF). However, there was no uniform law for the impact of slope gradients on TN loss.

The MAE, RMSE, NSE, and R^2^ of the effective mixing depth model in the numerical simulation of TN loss through SF from the sloping farmland in the Miyun Reservoir were 0.95 mg/L, 1.54 mg/L, 0.74, and 0.831, respectively. The same values of the convection-dispersion equation in simulating the change in TN concentration carried by the SSF were 24.99 mg/L, 39.25 mg/L, 0.81, and 0.811, respectively, which overall indicated good simulation results.

## Figures and Tables

**Figure 1 plants-12-02042-f001:**
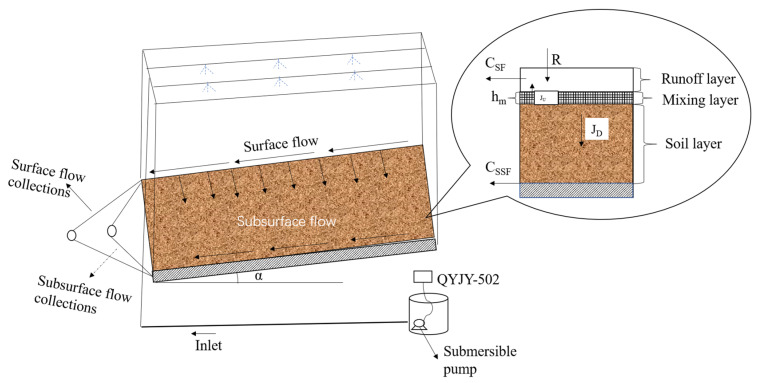
Schematic diagram of experimental structure. Note: R is rainfall intensity, mm/h; CSF is concentration of solute in surface flow, mg/L; CSSF is concentration of solute in subsurface flow, mg/L; hm is effective mixing depth, cm; JU and JD are upward and downward solute flux from soil layer, respectively, mg/cm·min; α is depth gradient, (°).

**Figure 2 plants-12-02042-f002:**
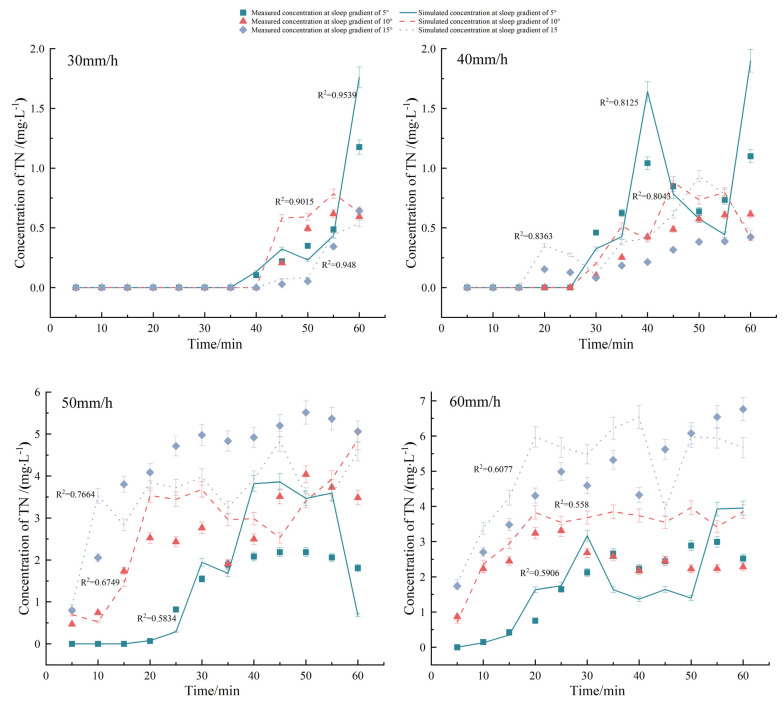
Comparison of measured and simulated concentrations of TN loss through surface flow. The error bars refer to the standard deviation.

**Figure 3 plants-12-02042-f003:**
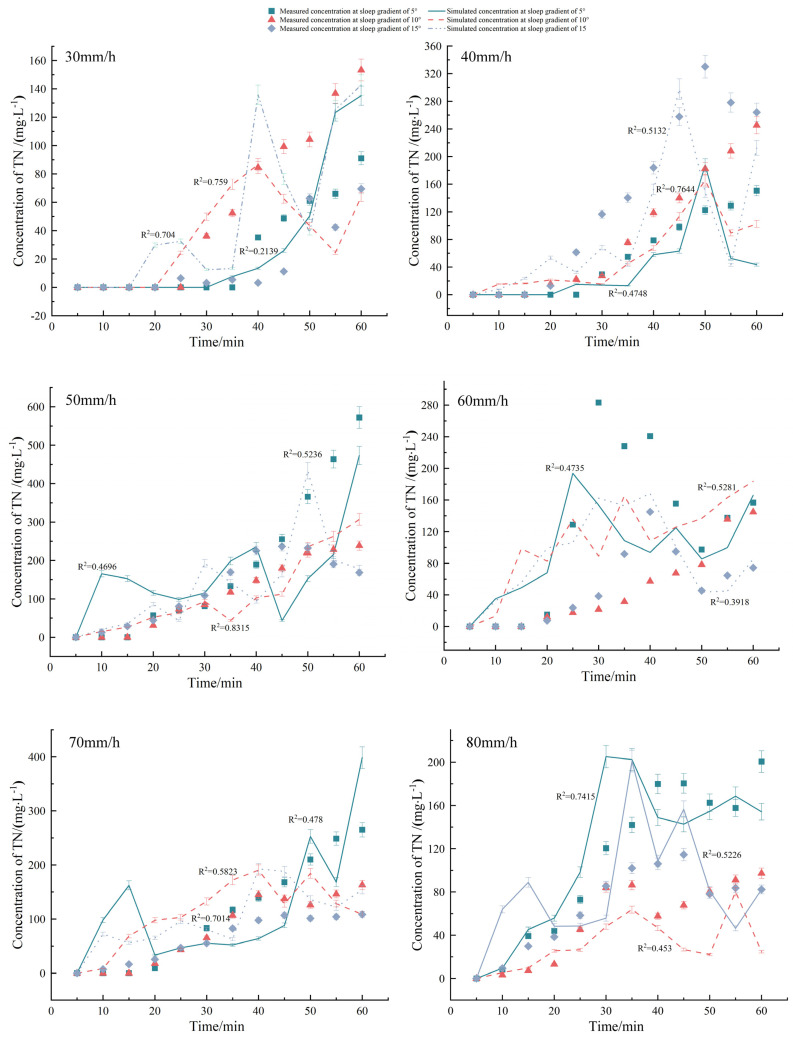
Comparison of measured and simulated concentration of TN loss through subsurface flow. The error bars refer to the standard deviation.

**Figure 4 plants-12-02042-f004:**
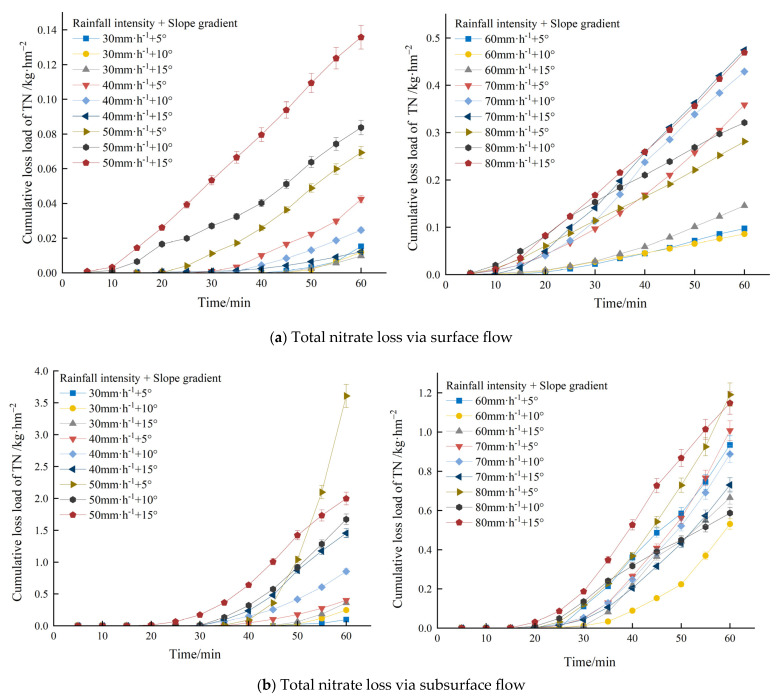
Cumulative loss loads of TN under different treatments.

**Figure 5 plants-12-02042-f005:**
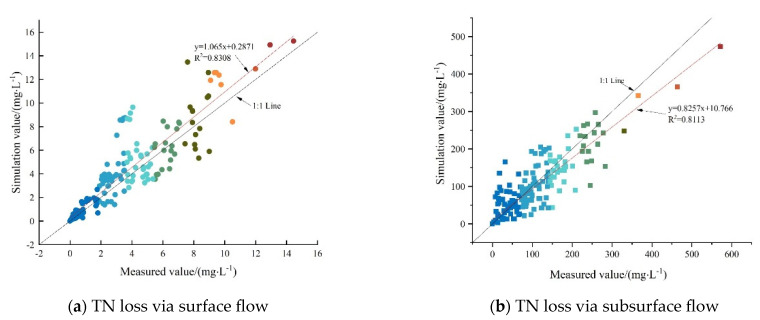
Scatter diagram of simulated and measured TN loss concentrations. (**a**). TN loss via surface flow (**b**). TN loss via subsurface flow.

**Table 1 plants-12-02042-t001:** Parameters for numerical simulations.

Parameters	*ρs*/ (g·cm^−3^)	θ*s*/ (cm^3^·cm^−3^)	*k*/ (cm^3^·g^−1^)	*D_L_*/ cm	*Dw*/ (cm^2^·min^−1^)	θ*r*/ (cm^3^·cm^−3^)
Surface	1.16	0.424	0.77	-	-	-
Subsurface	1.16	0.424	0.77	1.19	0.03	0.0348

Note: *ρs* is soil bulk density, *θs* is saturated water content, *θr* is residual saturated water content, *k* is the soil adsorption rate, *D_L_* is longitudinal dispersity of TN, *D_W_* is molecular diffusion coefficient in free water of TN.

**Table 2 plants-12-02042-t002:** Total nitrate loss concentrations and loads under different treatments.

Rainfall Intensity/(mm·h^−1^)	Slope Gradient/(°)	Mean Concentration ± Standard Deviation/(mg·L^−1^)		Mean Load ± Standard Deviation (kg/ha)	
		Surface Flow	Subsurface Flow	Surface Flow	Subsurface Flow
30	5	0.195 ± 0.350b	21.483 ± 1.100a	0.016 ± 0.002b	0.098 ± 0.002a
	10	0.159 ± 0.255b	27.898 ± 3.055a	0.014 ± 0.004b	0.247 ± 0.011a
	15	0.089 ± 0.200b	56.868 ± 5.282a	0.011 ± 0.005b	0.359 ± 0.017a
40	5	0.454 ± 0.436b	56.421 ± 4.478a	0.041 ± 0.006b	0.405 ± 0.077a
	10	0.255 ± 0.267b	87.932 ± 1.586a	0.025 ± 0.005b	0.857 ± 0.004a
	15	0.189 ± 0.157b	137.715 ± 9.576a	0.053 ± 0.071b	1.452 ± 0.122a
50	5	1.221 ± 0.961b	180.013 ± 14.808a	0.067 ± 0.013b	3.586 ± 0.093a
	10	2.487 ± 1.132b	108.718 ± 4.322a	0.083 ± 0.008b	1.648 ± 0.022a
	15	4.277 ± 1.442b	124.464 ± 1.938a	0.151 ± 0.013b	1.950 ± 0.074a
60	5	1.738 ± 1.108b	122.612 ± 2.594a	0.092 ± 0.015b	0.395 ± 0.040a
	10	2.394 ± 0.614b	47.245 ± 3.381a	0.101 ± 0.021b	0.528 ± 0.035a
	15	4.705 ± 1.520b	48.417 ± 1.994a	0.141 ± 0.018b	0.667 ± 0.023a
70	5	4.236 ± 1.983b	107.244 ± 4.978a	0.356 ± 0.014b	1.017 ± 0.012a
	10	5.536 ± 2.145b	79.586 ± 1.619a	0.459 ± 0.032b	0.884 ± 0.030a
	15	9.395 ± 2.884b	63.044 ± 2.207a	0.455 ± 0.027b	0.739 ± 0.059a
80	5	3.134 ± 0.926b	110.316 ± 2.946a	0.263 ± 0.044b	1.163 ± 0.138a
	10	4.817 ± 1.742b	55.251 ± 3.036a	0.327 ± 0.022b	0.586 ± 0.002a
	15	7.894 ± 1.910b	65.977 ± 2.847a	0.458 ± 0.044b	1.118 ± 0.057a

Note: Different letters indicate significant difference at 0.05 level between surface and subsurface flow.

**Table 3 plants-12-02042-t003:** Precision evaluation of numerical simulations for loss process of TN.

Rainfall Intensity /(mm·h^−1^)	Slope Gradient /(°)	Surface			Subsurface		
		MAE	RMSE	NSE	MAE	RMSE	NSE
30	5	0.07	0.18	0.73	24.84	34.08	−0.13
	10	0.06	0.12	0.75	16.33	30.84	0.57
	15	0.02	0.05	0.95	23.90	31.26	0.68
40	5	0.18	0.31	0.45	31.35	45.98	0.32
	10	0.11	0.17	0.57	36.63	57.50	0.55
	15	0.17	0.24	−1.52	65.44	95.01	0.36
50	5	0.71	0.96	0.24	110.15	136.61	0.45
	10	0.70	0.80	0.45	32.14	40.39	0.79
	15	1.00	1.16	0.29	52.30	78.53	0.16
60	5	0.01	0.78	0.46	57.27	74.11	0.40
	10	0.92	1.08	−1.34	61.27	72.38	−1.20
	15	0.80	0.96	0.56	42.38	57.38	−0.67
70	5	1.06	1.24	0.57	65.97	81.48	0.27
	10	1.14	1.48	0.48	44.55	51.98	0.30
	15	1.74	2.17	−0.31	42.76	50.05	−0.52
80	5	2.06	2.73	−0.68	26.93	36.89	0.72
	10	2.10	2.43	−1.12	23.99	32.74	0.17
	15	1.42	1.57	0.27	28.88	41.33	−0.28
All treatments		0.95	1.54	0.74	24.99	39.25	0.81

## Data Availability

Not applicable.

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
