# Peer review of "Laws Governing Nitrogen Loss and Its Numerical Simulation in the Sloping Farmland of the Miyun Reservoir"

_plants, 2023, doi:10.3390/plants12102042_

Round 1

Reviewer 1 Report

The reviewed manuscript is interesting and concerns the current issue of nitrogen losses in intensive agricultural production using crop irrigation.

Key words. is “model” I propose add “nitrogen simulation model”

Introduction

Page 1

Authors should explain in more detail why soil erosion threatens food security

Page 2

Please add point past [8,9}

Please changed “The N loss along with SF generally increases” with “The N loss along with SF significantly

Please detaily explain “Furthermore, the rainfall intensity generally impacts SF more than SSF.” increases”

Page 3 Material and Methods

Obligatory Authors should change chapter 2.1. with chapter 2.2.. First, the research area should be characterized, and only then the methods used should be described.

Page 4

2.2. Test soil

Authors should obligatory add. information which methods used to determine the content of available forms macronutriens. “The pH was 6.33, while the contents of organic matter, TN, available phosphorus, and available potassium were 9.97 g/kg, 0.448 g/kg, 4.55 g/kg, and 45.9 g/kg, respectively [23]” Literature number 23 is poorly selected because it does not list the methods used to determine pH, TN, available forms P, K. Which method Authors used to determine pH??? In water, in CaCL2 or in 1MKCl. This is very important information because it will determine the model of nutrients leaching from the soil.

Page 7

Is “erosion intensity. and ultimatelyPlease remove the dot

Page 11

Figure 4. Please change kg/hm-2 with kg/ha. Authors should use SI unit.

Discussion

Of the 45 items of literature, only 7 are items from outside China. Please, especially in the discussion of the results, expand it with other European and American literature items.

Please, in the Literature chapter, correct all cited literature items in accordance with the requirements for authors

It is a pity that the authors focused on NT losses and did not take into account the forms of mineral nitrogen NH4+ and NO3-

Author Response

Response to Reviewer 1 Comments

Dear editor, thank you very much for handling our manuscript. We have fixed all the detailed comments of reviewer and responses to minor comments of reviewer are presented below. We are thankful for constructive comments, which greatly helped use to improve the paper. We have carefully read all comments and followed. Specific responses to questions and comments are given below. We have applied all changes in the original manuscript file.

The reviewed manuscript is interesting and concerns the current issue of nitrogen losses in intensive agricultural production using crop irrigation.

Point 1. Key words. is “model” I propose add “nitrogen simulation model”

Response 1:We changed it as “nitrogen simulation model’’

Introduction

Point 2. Page 1

Authors should explain in more detail why soil erosion threatens food security

Response 2:We revised it.

Point 3. Page 2

Please add point past [8,9}

Please changed “The N loss along with SF generally increases” with “The N loss along with SF significantly

Response 3:We revised it.

Point 4. Please detaily explain “Furthermore, the rainfall intensity generally impacts SF more than SSF.” increases”

Response 4:Thank you. We defined the reason.

Point 5. Page 3 Material and Methods

Obligatory Authors should change chapter 2.1. with chapter 2.2.. First, the research area should be characterized, and only then the methods used should be described.

Response 5:We revised it. First, we characterized the research area and methods.

Point 6.Page 4

2.2. Test soil

Authors should obligatory add. information which methods used to determine the content of available forms macronutriens. “The pH was 6.33, while the contents of organic matter, TN, available phosphorus, and available potassium were 9.97 g/kg, 0.448 g/kg, 4.55 g/kg, and 45.9 g/kg, respectively [23]” Literature number 23 is poorly selected because it does not list the methods used to determine pH, TN, available forms P, K. Which method Authors used to determine pH??? In water, in CaCL2 or in 1MKCl. This is very important information because it will determine the model of nutrients leaching from the soil.

Response 6:We have revised it and written the methods of measurenments. 

Point 7. Page 7

Is “erosion intensity. and ultimately” Please remove the dot

Response 7:We corrected it. Thank you 

Point 8. Page 11

Figure 4. Please change kg/hm-2 with kg/ha. Authors should use SI unit.

Response 8:We corrected it. Thank you 

Point 9. Discussion

Of the 45 items of literature, only 7 are items from outside China. Please, especially in the discussion of the results, expand it with other European and American literature items.

Response 9:We corrected it. Thank you 

Point 10. Please, in the Literature chapter, correct all cited literature items in accordance with the requirements for authors

Response 10:We corrected it. Thank you 

Point 11. It is a pity that the authors focused on NT losses and did not take into account the forms of mineral nitrogen NH4+ and NO3-

Response 11:Thank for your suggestion. In our next study, we seperately measure NH4+ and NO3-. I this study we measure TN, because sample number was large. In addition funding was limited.

Reviewer 2 Report

In paragraph 3.1.1 the references have an inconsistency. The references [33] [21] are not Ahuja and [34] is about the HYDRUS and STANMOD applications.

 In paragraph 4.5 «However, the precision of the simulation results decreased with the increasing rainfall intensity, which was mainly because the measured initial loss concentration under great rainfall intensities was small and the TN concentration rapidly stabilized, thus causing an insignificant concentration increase in the actual measurement.» and «However, the R2 value fluctuated greatly (minimum of 0.21 and a maximum of 0.76), which was related to the large error between the measured values and simulated values.»

Also, in the Discussion paragraph at the end it summarizes «Nevertheless, the generation processes of both could not be simulated by the numerical model, resulting in the nutrient loss loading not being simulated and verified numerically. Moreover, there is a lack of coupling of the numerical model of the surface and underground nutrient loss process ».

With these references, the aim of the study for numerical model can give accurate estimates?

Author Response

Response to Reviewer 2 Comments

Dear editor, thank you very much for handling our manuscript. We have fixed all the detailed comments of reviewer and responses to minor comments of reviewer are presented below. We are thankful for constructive comments, which greatly helped use to improve the paper. We have carefully read all comments and followed. Specific responses to questions and comments are given below. We have applied all changes in the original manuscript file.

Point 1.In paragraph 3.1.1 the references have an inconsistency. The references [33] [21] are not Ahuja and [34] is about the HYDRUS and STANMOD applications.

Response 1: We have corrected these errors. Thank you very much,

Point 2. In paragraph 4.5 «However, the precision of the simulation results decreased with the increasing rainfall intensity, which was mainly because the measured initial loss concentration under great rainfall intensities was small and the TN concentration rapidly stabilized, thus causing an insignificant concentration increase in the actual measurement.» and «However, the R2 value fluctuated greatly (minimum of 0.21 and a maximum of 0.76), which was related to the large error between the measured values and simulated values.»

Also, in the Discussion paragraph at the end it summarizes «Nevertheless, the generation processes of both could not be simulated by the numerical model, resulting in the nutrient loss loading not being simulated and verified numerically. Moreover, there is a lack of coupling of the numerical model of the surface and underground nutrient loss process ».

With these references, the aim of the study for numerical model can give accurate estimates?

Response 2: Nice question. Thank you. Yes, numerical model provided accurate estimates. The different statistical indicators, including MAE, RMSE, NSE and R 2 in Figure 5 and Table 3 clearly shows simulation efficiency. The MAE, RMSE, NSE R 2 value are in acceptable range. In above references statements, we have defined factors affecting accurate estimatation. With increasing rainfall intensity, we can see that RMSEs values are increasing. Overal results show that thenumerical model can give accurate estimates. The simulated values of the TN concentration migrating with SF at small rainfall intensities was consistent with the measured values, with the determination coefficient (R2) reaching 0.9539, 0.9015, and 0.9480, respectively. The Nash-Suttcliffe efficient NSE were 0.73, 0.75, and 0.95, respectively, at the slope gradients of 5°, 10°, and 15°, with a rainfall intensity of 30 mm/h.

Reviewer 3 Report

Thank you for the opportunity to review this paper.

Nitrogen losses at the level of agricultural land represent a serious problem, especially at the level of agricultural land on the slope, where there is a reduction in agricultural productivity.

The Materials and Methods chapter presents the study methods proposed by the authors, methods that were chosen correctly and in accordance with the purpose of the work. Numerical simulation of the underground migration process and nitrogen losses was carried out in combination with the convective-dispersion mathematical model.

The obtained results are amply presented, the statistical analysis of the obtained data is well documented through numerous figures and tables. The manuscript is well scientifically documented.  The Discussions chapter reports its own results to the existing data in the literature. As the authors also explained, this paper provides a reference for the study of the mechanism of nitrogen migration and loss in sloping agricultural lands. Conclusions established are the result of the research activity.

I recommend this paper be accepted and published in this journal.

Author Response

Response to Reviewer 3 Comments

Dear editor, thank you very much for reviewing our manuscript. This paper provides a reference for the study of the mechanism of nitrogen migration and loss in sloping agricultural lands. We’ll keep working hard and persist to the end. We look forward to making contributions to the prevention and control of nitrogen loss in sloping farmland.
